# α-Ketoglutaric Acid-Modified Carbonate Apatite Enhances Cellular Uptake and Cytotoxicity of a Raf-Kinase Inhibitor in Breast Cancer Cells through Inhibition of MAPK and PI-3 Kinase Pathways

**DOI:** 10.3390/biomedicines7010004

**Published:** 2019-01-03

**Authors:** Sultana Mehbuba Hossain, Jayalaxmi Shetty, Kyi Kyi Tha, Ezharul Hoque Chowdhury

**Affiliations:** 1Jeffrey Cheah School of Medicine and Health Sciences, Monash University Malaysia, Jalan Lagoon Selatan, Bandar Sunway, 47500 Subang Jaya, Malaysia; hossainmishu_1990@yahoo.com (S.M.H.); jayashetty21@gmail.com (J.S.); Tha.Kyi.Kyi@monash.edu (K.K.T.); 2Health & Wellbeing Cluster, Global Asia in the 21st Century (GA21) Platform, 47500 Subang Jaya, Malaysia

**Keywords:** AZ628, carbonate apatite nanoparticles, α-ketoglutaric acid, Raf-kinase inhibitor, breast cancer, cytotoxicity, MAPK, PI-3 kinase, tumor regression

## Abstract

AZ628 is a hydrophobic Raf-kinase inhibitor (rapidly accelerated fibrosarcoma) currently in clinical trial of various cancer. The physicochemical properties of hydrophobic drugs that affect the drug-particle interactions and cause aggregation of drugs and particles might be the key aspect to impede effective drug delivery. Retaining smaller particle size is the prerequisite to overcome the opsonization and improve cytotoxicity in the targeted region. Carbonate apatite (CA), an attractive biodegradable vector, has been used to carry both hydrophilic and hydrophobic drugs and release the payloads inside the cells following endocytosis. We incorporated AZ628 into CA and also modified it with α-ketoglutaric acid (α-KA) for reducing particle growth kinetics and increasing total surface area to improve the delivery of AZ628 by enhancing cellular uptake by breast cancer cells. AZ628-loaded nanoparticles of CA and α-KA-modified CA (α-KAMCA) were synthesized and evaluated in MCF-7 and 4T1 cell lines by measuring cytotoxicity and cellular uptake analysis. HPLC (high-performance liquid chromatography) assay was performed to quantify the binding affinity of the nanocarriers towards the drug. Western blot analysis was done to see the activation and expression levels of Akt, MAPK (mitogen-activated protein kinase) pathways and Caspase-3. Zetasizer was used to measure the particle size along with the surface charge. α-KAMCA showed almost 88% encapsulation efficacy for AZ628 with around 21% enhanced cellular uptake of the drug in two different breast cancer cell lines. These findings suggest that α-KAMCA could be a promising therapeutic tool to carry AZ628 for breast cancer treatment.

## 1. Introduction

Breast cancer is the most commonly detected cancer [1] in women with 1,700,000 new cases worldwide. Almost 883,000 new cases were identified in developing countries along with 794,000 new cases in developed countries, making the disease the fifth common reason for women’s death per year [2,3,4]. The morbidity and mortality rate are high due to the distant metastasis which involves cancer cell detachment from the primary site of the tumor, entrance into the systemic circulation and tumor cell proliferation in the other organs parenchyma [5]. Hence it is important to deliver the drugs in a targeted manner for the suppression of breast cancer cells metastasis or invasiveness.

The Raf/MEK/ERK signalling pathway (Raf = rapidly accelerated fibrosarcoma, MEK = MAPK/ERK kinase, ERK = extracellular signal–regulated kinases) is recognized as the mitogen-activated protein kinase (MAPK) cascade comprising guanosine-nucleotide-binding protein RAS (GTPase) which stimulates Raf-family proteins including A-Raf, b-Raf and c-Raf/Raf1 [6]. This signalling cascade controls cell proliferation, differentiation, metastasis, survival, and angiogenesis. RKIP (Raf-kinase inhibitor protein) is a master modulator of the Raf/MEK/ERK signalling pathway, which can interfere with the Raf-1 facilitated phosphorylation and activation by interrupting the interaction between the MAP-kinase and the MAP-kinase activated extracellular signal-regulated kinase (ERK) [1,5,6,7]. However, RKIP expression is highly reduced in lymph node metastasis of breast cancer [7].

The impaired outcome of the Raf/MEK/ERK signalling cascade may cause the incidence of different types of tumorigenesis [6]. As such, Raf-1 controls the anti-apoptotic transcription factor, nuclear factor-κB as well as caspase-8 activation. Again, overexpression of Raf-1 may cause multi-drug resistance phenotype [8]. In contrast, b-Raf is more commonly mutated than A-Raf and c-Raf [9] in a human tumor, promoting tumour cell proliferation and survival by elevating kinase activity and stimulating downstream MEK–ERK signaling [10]. Hence it is a crucial treatment strategy to inhibit activation of b-Raf mutation, resulting in downregulation of the MEK kinase and preventing the cancer cell growth.

Hydrophobic in nature, AZ628 is a type II selective and potent pan-Raf kinase inhibitor under clinical pipeline by AstraZeneca. It impedes anchorage-dependent and -independent growth, resulting in the arrest of the cell-cycle, and induces apoptosis in different cancer cell lines [9]. By inhibiting the activity of preactivated b-Raf, b-Raf V600E and c-Raf, it stops MEK activation and sustains sensitivity to the MEK inhibitor, thus causing effective inhibition of cancer cell proliferation [10,11].

The targeted delivery of a hydrophobic anticancer drug is a challenging process due to its solubility issue. Owing to hydrophobicity, the chemotherapeutics are not readily soluble in the aqueous environment of the body and tissue fluids. Moreover, the hydrophobic anticancer drugs are prone to aggregation upon intravenous administration, leading to embolism and local cytotoxicity [12]. Nanoparticles (NPs) present an auspicious approach to address the lipophilicity issue of a hydrophobic drug and can improve drug accumulation in tumor sites by altering the biodistribution and pharmacokinetic properties [4,13,14,15]. Since NPs are substantially larger in size than the anticancer drugs, they cannot cross the tight junctions between endothelial cells of the healthy blood vessel. This phenomenon generally prohibits the NPs from carrying the chemotherapeutics inside the healthy tissue. The tumour site is, on the other hand, composed of the leaky vasculature and reduced number of weakened lymphatic drainage, thus enabling the right sized NPs to enter the site with long-term accumulation. This phenomenon, known as the enhanced permeability and retention (EPR) effect, is the basic concept for a passive tumour targeting [12].

Broadly speaking, NPs can be grouped into organic and inorganic nanomaterials. Depending on the physicochemical properties, inorganic nanomaterials are usually prefered over organic ones [16], mostly because of the easier formulation process, controlled particle size distribution, targeted delivery of the drug, stability and rapid release of the payload [17,18,19].

Carbonate apatite (CA) Ca_10_(PO_4_)_6-*x*_(CO_3_)*_x_*(OH)_2,_ is a smart non-viral pH-sensitive inorganic carrier owing to its biodegradability, heterogeneous surface charge, and limited crystal growth. CA NPs possess PO_4_^3−^ and HCO_3_^−^-rich and Ca^2+^-rich domains in their outer surfaces that provide a broader scale of binding opportunity for both positively and negatively charged drugs through ionic interactions. From the systemic circulation, the drug-loaded CA NPs can enter into the cancer cell via endocytosis, resulting in particle degradation and discharge of the encapsulated drug inside the endosomal acidic microenvironment [20,21]. However, CA NPs tend to form larger particles by aggregation process [22], which can be stabilized by the addition of serum albumin [23] or biotin-PEG [17].

Krebs cycle compounds such as sodium citrate, sodium succinate and α-ketoglutaric acid are natural, inexpensive, harmless and environment-friendly [24]. In our previous study, we found that the incorporation of citrate and succinate in CA NPs played a pivotal role in modulating the particle size, cellular uptake, amount of drug binding and cytotoxicity, depending on the number of carboxylic acid groups in the salt structures [4]. Here, α-ketoglutaric acid, an intermediate of Krebs cycle was used to transform the CA particles into α-ketoglutaric acid-modified CA (α-KAMCA) NPs. The effects of such modification on the particle size, drug-loading efficiency, and enhancement of cellular uptake and cytotoxicity of AZ628 were also examined.

## 2. Results

### 2.1. Characterization of α-KAMCA NPs

Synthesis of α-KAMCA NPs was carried out in an aqueous medium at different concentrations (1–16 mM) of α-KA salt and 4 mM of Ca^2+^ and confirmed by measuring turbidity at 320 nm in UV-VIS spectrophotometer. An increasing trend of turbidity was noted for α-KAMCA NPs with an increasing concentration of α-ketoglutaric acid, probably due to the increased number of particles and slow enlargement of the particle size (Figure 1A).

α-KAMCA NPs formulated with different concentrations of alpha-ketoglutaric acid salt were exposed to MCF-7 cell line to evaluate the probable toxic effect of the NPs. There was no significant decrease in cell viability with the use of α-KAMCA NPs formulated with different concentrations (1–16 mM) of α-ketoglutaric acid (Figure 1B) compared to CA NPs, indicating that α-KAMCA NPs do not cause any additional cytotoxicity to the cells.

In order to systematically validate the synthesis of α-KAMCA NPs, optical image analysis was performed to characterize the NPs. Optical microscopic images were taken immediately after α-KAMCA NPs were formed using different concentrations of α-ketoglutaric acid (Figure 2). The formation of NPs was almost unaffected in terms of size and number of aggregated particles when 1 mM and 2 mM of α-ketoglutaric acid were used with 4 mM Ca^2+^ (Figure 2C,D). Figure 2E, presents a uniformly distributed smaller particle size for the α-KAMCA NPs formulated with 4 mM concentration of alpha-ketoglutaric acid. However, a visible change of aggregated particle number and size was observed at a higher concentration (8–16 mM) of α-ketoglutaric acid (Figure 2F,G).

### 2.2. Size and Surface Charge

Nanoparticle size and surface charge are the most significant determinants that influence their physicochemical properties, optical properties, stability, release properties and cellular uptake [25,26,27]. The particle size (z-average) of CA NPs generated using 4 mM exogenous Ca^2+^ was around 428.4 ± 21.70 nm (Figure 1C). The average size (z-average) and zeta potential of formulated α-KAMCA NPs were measured using different concentrations of α-ketoglutaric acid. The concentration-independent particle size was noticed with α-KAMCA NPs. Alpha-ketoglutaric acid added at 1, 2, 4, 8 and 16 mM concentrations along with 4 mM Ca^2+^ generated the particles of 467 ± 27.29, 415.9 ± 0.78, 291.5 ± 10.60, 382.9 ± 5.45 and 432.6 ± 5.30 nm, respectively (Figure 1C). The optical image analysis of NPs also proved that the mixing of α-ketoglutaric acid and Ca^2+^ at 1:1 ratio to formulate the α-KAMCA NPs was optimum to achieve a controlled particle size with an adequate number of particles (Figure 2E).

AZ628, a hydrophobic drug, showed a tendency to increase the particle size of both CA NPs and α-KAMCA NPs upon complexation. The average size (z-average) of AZ628-loaded CA NPs was 456 ± 5.66, 483.5 ± 7.78, 535.5 ± 3.53 and 591.5 ± 1.98 nm when 1, 10, 100 and 1000 nM concentrations of AZ628 were used, respectively (Figure 3A). AZ628-loaded α-KAMCA NPs presented drug concentration-dependent particle size enlargement with 1, 10, 100 and 1000 nM drug concentrations, which lead to particles of 366.5 ± 4.95, 391 ± 2.83, 430.5 ± 26.16 and 478.7 ± 6.50 nm, respectively (Figure 3B).

No significant change in the surface charge ranging from −9.67 to −12.2 mV was observed by the addition of α-ketoglutaric acid to form α-KAMCA NPs (Figure 3C). The surface potential remained the same for the AZ628-incorporated CA and α-KAMCA NPs (Figure 3D) suggesting that AZ628 does not interfere with the net surface charge modification of the apatite-based NPs.

### 2.3. Dynamic Light Scattering (DLS)

DLS technique is the most popular tool to measure the diameter of particles in suspension [28,29,30] and the stability of the formulation [31]. Typically, it measures particle size distribution by analyzing the intensity of the scattered light of the sample particles with some limitations of poor accuracy for the larger sized particles and highly polydisperse samples [31,32,33].

CA and α-KAMCA NPs were subjected to DLS to quantify the particle size based on the % intensity and polydispersity index (PDI) value, which is a standard tool to determine the uniformity of the particle size. For CA NPs containing a colloidal suspension (Figure 4A), 718.3 nm particle size was recorded with a PDI value of 0.553. The addition of 1 mM and 2 mM α-ketoglutaric acid with CA NPs controlled the particle sizes to 407.3 nm and 504.4 nm (62.1%) with reduced PDI values of 0.566 and 0.433, respectively (Figure 4B,C). Interestingly, α-KAMCA formulated with 4 mM α-ketoglutaric acid and 4 mM calcium possessed 289.9 nm particle size with lower PDI value of 0.322 (Figure 4D), characterizing a monodisperse distribution with a reduced particle size. However, a further increase in the α-ketoglutaric acid concentration increased the particle size/intensity along with the PDI value (Figure 4E,F), indicating the agglomeration tendency of α-KAMCA NPs.

### 2.4. AZ628 Encapsulation in Inorganic-Apatite NPs

Drug encapsulation in NPs enhances the solubility profile of a drug, stabilizes it, minimizes its toxicities and delivers it to the targeted site by EPR effect [34]. CA and α-KAMCA NPs exhibited relatively high AZ628-loading capacities (almost 82–88% drug loading efficiency) (Figure 5). AZ628 revealed a concentration-independent drug loading into inorganic-apatite based CA and α-KMCA NPs. At 60 µM drug concentration, the amounts of AZ628 incorporated into CA and α-KAMCA NPs are ~49.73 and ~51.63 µM (Figure 5A), representing almost 82.87% and 86.05% encapsulation efficiency (Figure 5B) of the initial concentration of the drug, and 14.97% and 18.68% drug loading capacity (Table 1), respectively. At a higher concentration (100 µM), the amounts of AZ628 incorporated into CA and α-KAMCA NPs are ~87.65 µM (Figure 5C) which is almost 88% encapsulation of the initial dose (Figure 5D). Interestingly, α-KAMCA NPs possessed 32.28% drug loading capacity for 100 μM of AZ628, while CA NPs demonstrated 26.96% drug loading capacity (Table 1), which clearly indicates that α-KAMCA NPs could successfully entrap up to 5% more drug than CA NPs per unit mass of the particles.

### 2.5. Anticancer Effect of Only AZ628 and AZ628-Loaded CA and α-KAMCA NPs

The human breast cancer cell line MCF-7 was used for the in vitro experiment due to its hormone sensitivity and oestrogen receptor (ER) expression [35]. The murine mammary adenocarcinoma 4T1 cell line was also selected to mimic the breast cancer model for its resemblance to human triple negative breast cancer [36] and also an animal model for stage IV human breast cancer. Thus, the enhanced therapeutic efficacy of AZ628-loaded CA NPs and AZ628-loaded α-KAMCA NPs was evaluated by MTT assay on MCF-7 and 4T1 and compared with the effect of free AZ628 on inhibition of MCF-7 and 4T1 cell proliferation at 48 h of treatment. The cytotoxicity of free AZ628, AZ628-loaded CA and AZ628-loaded α-KAMCA NPs are presented in Figure 6, with cell viability for CA and α-KAMCA NPs observed to be 92% and 94%, respectively in MCF-7 cell line (Figure 6A,B). The % cytotoxicity of AZ628-loaded α-KAMCA NPs was much higher than that of the similar dose of AZ628-loaded CA NPs, and only AZ628. At 100 pM concentration, only AZ628 caused 19.58% and 11.76% cell death and at 1µM concentration 27.9% and 36.71% cell death in MCF-7 and 4T1 cell line respectively (Figure 6A–D). At 1 µM initial concentration of AZ628, drug-loaded α-KAMCA NPs caused 42.74% (*p* < 0.0001) and 33.19% (*p* < 0.0001) cell viability (Figure 6B,D) and drug-loaded CA NPs showed 51.19% (*p* < 0.0001) and 45.74% (*p* < 0.01) cell viability (Figure 6A,C) in MCF-7 and 4T1 cell line, respectively. The results showed that AZ628-loaded α-KAMCA NPs possessed almost 9% and 12% more cytotoxicity compared to AZ628-loaded CA NPs, and 30% more cytotoxicity compared to the same concentration of free drug in MCF-7 and 4T1 cells, respectively.

The time-dependent anticancer efficacy also was evaluated for free AZ628, AZ628-loaded CA NPs and AZ628-loaded α-KAMCA NPs on MCF-7 at 24 h of treatment (Figure 7). At 1 µM concentration of AZ628, drug-loaded α-KAMCA NPs caused almost 57% (*p* < 0.001) cell viability (Figure 7B) and drug-loaded CA NPs showed almost 61% (*p* < 0.01) cell viability (Figure 7A) in MCF-7 cell line, representing approximately 18% and 14% more cytotoxicity compared to the same concentration of free AZ628, respectively.

The augmentation of % cytotoxicity for only AZ628 and AZ628-loaded CA, AZ628-loaded α-KAMCA NPs in MCF-7 and 4T1 cells has been summarized in Table 2 and Table 3. AZ628-loaded CA NPs and AZ628-loaded α-KAMCA NPs revealed the drug concentration-dependent augmentation in cytotoxicity in the MCF-7 cell line, whereas in the 4T1 cell line both types of NPs with loaded drugs possessed concentration-independent enhanced cytotoxicity. The substantial increase in cytotoxicity was 21% (*p* < 0.0001) and 23% (*p* < 0.0001) noticed with 1 µM concentration of AZ628 initially used to load into α-KAMCA NPs, compared to the free drug in MCF-7 and 4T1 cell line, respectively. AZ628-loaded CA NPs revealed 10% (*p* < 0.0001) at 1 µM strength and 9% (*p* < 0.01) greater cytotoxicity at 100 nM of AZ628 in MCF-7 cells and 4T1 cell lines, respectively.

The therapeutic efficacy of AZ628-bound NPs was scaled up by using different concentrations of AZ628 ranging from 100 pM to 1 µM in MCF-7 and 4T1 cells. The entire treatment group presented a dose-dependent cytotoxicity after 48 h of treatment. The IC50 in MCF-7 cells for AZ628-loaded CA NPs was found at 1 µM, ensuring almost 50% cellular death compared to the untreated cells, whereas AZ628-loaded α-KAMCA NPs caused around 50% cell death at 10 nM concentration of AZ628 (Figure 6A,B). In 4T1 cell line, AZ628-loaded α-KAMCA NPs revealed 50% cell death at 100 nM concentration of the drug (Figure 6D); however, the IC_50_ value remained same for AZ628-loaded CA NPs as in MCF-7 cell line (Figure 6C).

### 2.6. Cellular Uptake Study

Cellular uptake of AZ628, AZ628-loaded CA NPs and AZ628-loaded α-KAMCA NPs by MCF-7 and 4T1 cells was determined by HPLC analysis. As shown in Figure 8, apatite-based NPs had significantly increased cellular uptake compared to free AZ628 in both cell lines.

The cellular uptake of free AZ628 was 31.06% and 32.83% in MCF-7 cells for 40 and 100 µM concentration, respectively at 4 h of treatment. α-KAMCA-facilitated cellular uptake for AZ628 was ~40% and 58.97% (*p* < 0.0001) for 40 and 100 µM concentration of AZ628, respectively at 4 h of treatment in MCF-7 cell line; whereas CA NPs showed almost 34.38% and 48.97% (*p* < 0.001) cellular uptake for 40 and 100 µM concentration, respectively at 4 h of treatment (Figure 8A,B). α-KAMCA showed 17% and 25% more cellular uptake than AZ628-loaded CA and free AZ628 at 40 µM concentration, respectively. At higher concentration (100 µM), α-KAMCA NPs showed 10% and 34% higher cellular uptake than AZ628-loaded CA NPs and free AZ628, respectively.

Similarly, in 4T1 cell line, α-KAMCA NPs demonstrated 49% and 56.95% (*p* < 0.01) cellular uptake for AZ628, whereas CA NPs confirmed 32,66% and 46.1% (*p* < 0.05) cellular uptake for the drug at 40 and 100 µM concentration, respectively (Figure 8C,D).

2.7. pH-Responsive Degradability of Different NPs

To predict the pH-dependent release profile of the NPs and AZ628-loaded NPs, dissolution behavior of differently formulated NPs (in the absence or presence of the drug) was evaluated at different pH ranging from physiological blood pH 7.4 to late endosome pH 5.0 (Figure 9). The α-KAMCA NPs exhibited almost 58%, 68%, 83% and 97% dissolution within 5 min in pH 6.5, 6.0, 5.5 and 5.0, respectively (Figure 9), indicating that drugs would release at a faster rate at more acidic pHs through dissolution of α-KAMCA NPs, and that the modification of CA with α-ketoglutaric acid did not result in any significant alteration in the dissolution. Hence, like CA NPs, α-KAMCA NPs could release the drug very rapidly in an acidic tumor environment (endosomes) and discharge the payload into the cytoplasm.

### 2.8. Characterization of α-KAMCA NPs and CA NPs by FE-SEM

NPs morphology such as shape, size, functional group and organization plays an influential role in determining their physicochemical and functional properties, the cellular uptake, and drug binding and drug delivery to the targeted cells [37,38]. The morphology of α-KAMCA NPS was characterized by FE-SEM images. As seen in Figure 10A, α-KAMCA NPs had a cube-like shape with an average particle size being around 256–299 nm. The surface of the NPs possessed huge folds, providing a large surface area for the vast amount of drug binding. However, CA NPs owned a uniform spherical shape with comparatively less folding on the surface area than α-KAMCA NPs (Figure 10B), which accounts for the binding of a lower amount of the drug.

### 2.9. AZ628 Downregulates MAPK and Akt Expression and Inducces Caspase-3 Cleavage

MAP-kinase and Akt pathways play a promising role in cancer cell proliferation, differentiation and survival, whereas Caspase-3 is a crucial mediator of apoptosis; hence, the activation of these three pathways was investigated by using CA NPs, α-KAMCA NPs, AZ628, AZ628-loaded CA NPs and AZ628-loaded α-KAMCA NPs. Figure 11 showed AZ628, AZ628, AZ628-loaded CA NPs and AZ628-loaded α-KAMCA NPs enabled to knockdown MAP-kinase (ERK 1/2) and the Akt (Ser 473) pathway by remarkably inhibiting their phosphorylation. AZ628-loaded apatite-based NPs showed more Akt downregulation than the free AZ628, which may trigger apoptosis in the MCF-7 cell line. A similar pattern of downregulation of p-MAP kinase was observed on the Western blot analysis. It might be due to the high affinity of NPs for the drug, effective cellular uptake of the drug-loaded NPs, and enhanced intracellular drug release. The upregulation of Caspase-3 was noticed compared to the control, resulting in the induction of a cell apoptotic pathway. The terminal phase of apoptosis is usually promoted by the caspase-3 protein which can be triggered by either a death receptor-dependent or mitochondrion-dependent signaling pathway.

### 2.10. Stability of CA and α-KAMCA NPs

A successful drug delivery nanocarrier requires stability during storage and inside a biological system. A biological condition was maintained during a stability test depending on the final application. The ultimate fate of CA and α-KAMCA NPs was to deliver the drug inside the tumor microenvironment while being stable in a normal physiological condition. To ensure the normal physiological condition, the incubation temperature and the pH of the media were maintained at 37 °C and pH 7.4, respectively, during the process. CA and α-KAMCA NPs showed an increase in turbidity with the duration of time (Figure 12), indicating that the particle aggregation was drastically reduced through interactions with serum proteins. α-KAMCA NPs showed a gradual rise in turbidity while CA NPs possessed a sharp increase in turbidity from 0.5 h to 8 h after formulation. The turbidity at 24 h (day 1), 48 h (day 2) and 72 h (day 3) exhibited a slow increase which is indicative of an increased particle number as particle aggregates interact with serum proteins, thereby suggesting that the NPs are stable at a physiological body environment.

## 3. Materials and Methods

### 3.1. Materials and Chemicals

DMEM (Dulbecco’s modified eagle medium) powder was procured from Gibco by Life Technology (Thermo Fisher Scientific, Waltham, MA, USA). Sodium bicarbonate (NaHCO_3_), calcium chloride (CaCl_2_.2H_2_O), and alpha-ketoglutaric acid disodium salt hydrate salts were purchased from Sigma-Aldrich (St Louis, MO, USA). Anti-cancer drug AZ628, DMSO (Dimethyl sulphoxide) and thiazolyl blue tetrazolium bromide (MTT) were obtained from Sigma–Aldrich (St Louis, MO, USA). DMEM-high glucose liquid media, FBS (Fetal Bovine Serum), TrypLE Express and penicillin-streptomycin were acquired from Sigma–Aldrich (St Louis, MO, USA). Acetonitrile (ACN) was from Fischer Scientific (Loughborough, UK). Bradford 1× dye reagent was purchased from Bio-rad Laboratories (Hercules, CA, USA).

### 3.2. Fabrication and Turbidity Measurement of CA and α-KAMCA NPs

DMEM-buffered solution was prepared by dissolving 0.675 g DMEM powder with 0.185 g NaHCO_3_ in 50 mL Milli Q water and 1 N hydrochloric acid was used to fix the final pH of the solution to 7.4 [21,39]. CA NPs was generated by mixing 4 mM of calcium chloride dihydrate in 1 mL of freshly prepared DMEM solution and the final mixture was incubated at 37 °C for 30 min. For optimizing the particle formation, alpha-ketoglutaric acid disodium salt hydrate at 1, 2, 4, 8 and 16 mM was dissolved separately along with 4 mM exogenous Ca^2+^ in 1 mL of DMEM media and incubated all the combination at 37 °C for 30 min to formulate α-KAMCA NPs. After 30 min incubation, the turbidity of CA and α-KAMCA NPs were assessed at 320 nm wavelength through a UV-visible spectrophotometer (Jasco, Oklahoma, OK, USA).

### 3.3. Turbidity Measurement of Alpha-Ketoglutaric Acid Salt

The turbidity of alpha-ketoglutaric acid salt dissolved in DMEM media was checked to analyze any possible interference of the salt with the turbidity of the formulated α-KAMCA NPs in 320 nm wavelength. Thus, α-ketoglutaric acid salt at 1, 2, 4, 8 and 16 mM was added to 1 mL DMEM solution and incubated at 37 °C temperature for 30 min. All other conditions for the particle formation were maintained in the same manner and the turbidity of the salt in the DMEM media was measured.

### 3.4. Optical Images of α-KAMCA Particle Formation

The images of aggregated particles were taken by Olympus Fluorescence Microscope IX81 (Shinjuku, Tokyo, Japan) with 10× magnification at a scale bar of 50 µm. α-KAMCA particles were formed by mixing alpha-ketoglutaric acid salt at 0, 1, 2, 4, 8 and 16 mM concentrations with 4 mM Ca^2+^ and incubating for 30 min at 37 °C. After the particle formation, all the particle suspensions were transferred to a 24-well plate and optical images were taken immediately.

### 3.5. Estimation of Drug Encapsulation Efficacy

The high-performance liquid chromatography method was performed to estimate the concentration of the drug encapsulated in the NPs. The standard curve was prepared by plotting the concentration of the drug (AZ628) vs. the peak area. The drug was used at 0, 20, 40, 60, 80 and 100 µM concentrations to get the peak area. AZ628 at 60 and 100 µM concentration was added with 4 mM exogenous calcium in 1 mL DMEM media to create AZ628-bound CA NPs. Similarly, AZ628 at 60 and 100 µM concentration was mixed with 4 mM exogenous Ca^2+^ and 4 mM alpha-ketoglutaric acid salt in 1 mL DMEM solution to produce AZ628-loaded α-KAMCA NPs. The mixture of salts and drug was then incubated at 37 °C for half an hour. Next, the drug-particle suspension was centrifuged at 13,000 rpm for 30 min at 4 °C using a Refrigerated Bench-Top Microcentrifuge (Eppendrof, Hamburg, Germany). After the particle precipitation, the supernatant was collected and the amount of the drug present in the supernatant was checked by Agilent chemostation software attached with HPLC (Agilent, Santa Clara, CA, USA). To perform this experiment, zorbax C18 column (4.6 × 150 mm, Agilent, Santa Clara, CA, USA) was used and the mobile phase was ACN and Milli Q water in a proportion of 82.5:17.5 (*v*/*v*), pumped at a constant flow rate of 1.0 mL/min. Quantification was performed at DAD (Diode Array Detector) wavelength of 254 nm.

The standard curve (*y* = 8.0079*x* + 19.214, R^2^ = 0.9981) was used to calculate the concentrations of AZ628 present in the supernatant.
$$\left[N\right]\;unbound\;drug\;conc.=\frac{\left[P\right]\;peak\;area\;of\;supernatant-19.214}{8.0779}$$
[M] drug bound with NP=[M]initial drug conc.−[N] unbound drug conc.

The % drug encapsulation efficiency was calculated using the following formula:$$\%\;Encapsulation\;efficiency=\frac{\left[M\right]\;drug\;bound\;with\;NP}{\left[M\right]initial\;drug\;conc.}\times 100\%$$
where ‘[*M*] *drug bound with NP*’ is the concentration of AZ628 entrapped in NPs calculated from the standard curve and ‘[*M*] *initial drug conc.*’ is the total concentration of AZ628 used to run HPLC (or initially mixed for the preparation of AZ628-loaded NPs formulations), “[*N*] *unbound drug conc.*” is the concentration of free AZ628 in the supernatant (the amount of drug that is not bound with the NPs), “[*P*] *peak area of supernatant*” is the area of unbound AZ628 in the supernatant detected by HPLC.
$$\%\;Drug\;loading\;capacity=\frac{Mass\;of\;AZ628\;bound\;to\;NPs}{Mass\;of\;NPs\;recovered}\times 100\%$$
where “*Mass of AZ628 bound to NPs*” is the amount of AZ628 entrapped in NPs (μg/mL) and “*Mass of NPs recovered*” is the number of NPs collected after centrifugation (μg/mL). The experiments were performed in triplicate and presented as average ± SD (standard deviation).

### 3.6. Size and Surface Charge Measurement

For the size and surface charge measurement, AZ628-loaded α-KAMCA NPs were generated by adding the drug at 1, 10, 100 and 1000 nM concentrations, together with 4 mM Ca^2+^ and 4 mM alpha-ketoglutaric acid salt to 1 mL of freshly prepared DMEM. AZ628 in 1 nM to 1 µM concentration with 4 mM Ca^2+^ was added to 1 mL freshly prepared DMEM media to formulate AZ628-loaded CA NPs. In a similar way, drug-free α-KAMCA particles were prepared by mixing alpha-ketoglutaric acid salt at 1 mM to 16 mM strength with 4 mM exogenous Ca^2+^ to 1 mL of freshly prepared DMEM solution. CA NPs was formulated by mixing 4 mM of Ca^2+^ in 1 mL of freshly prepared DMEM solution. All the mixtures of salts were incubated at 37 °C for 30 min, after which 10% FBS was added to all the preparations. The experiment was performed without any dilution of the samples. During measurement, all the preparations were kept in the 4 °C ice chiller. Malvern Nano Zetasizer (Malvern, Worcestershire, UK) was used to measure the particle size (z-average) in diameter, the particle size by intensity (dynamic light scattering) and the zeta potential.

### 3.7. Culture and Seeding

The human breast cancer cell line, MCF-7 cells, and the mouse breast cancer cell line, 4T1 cells, were cultured in two separate 25 cm^2^ flasks with a complete DMEM (cDMEM) media (pH 7.4) containing 10% FBS, penicillin and streptomycin antibiotic and the flasks were placed in a humidified incubator at 37 °C with 5% CO_2_. Both cell lines were collected from the exponential growth phase, subjected to trypsinization process, following repeated washing through centrifugation steps and seeded on a 24-well plate (Greiner, Frickenhause, Germany). Each of the wells contained approximately 50,000 cells and was subjected to overnight incubation prior to treatments.

### 3.8. Fabrication of NPs and AZ628-Loaded NPs and Cell Treatment

DMEM solution was freshly prepared according to the above-mentioned protocol and the pH of the final media was adjusted to 7.4 using 1 N HCl. AZ628 at 0.1, 1, 10, 100 and 1000 nM concentrations was added along with 4 mM calcium and 4 mM alpha-ketoglutaric acid salt to 1 mL of filtered DMEM solution. Likewise, AZ628 in 0.1, 1, 10, 100 and 1000 nM strength was added with 4 mM Ca^2+^ to 1 mL filtered DMEM media. These mixtures were then incubated for 30 min at 37 °C to generate AZ628-loaded α-KAMCA NPs and AZ628-incorporated CA NPs. The same concentrations of AZ628 were added to 1 mL DMEM solution and incubated in the same manner to prepare free drug solutions as a control. Drug-free α-KAMCA NPs were also fabricated by the addition of alpha-ketoglutaric acid salt at 1 mM to 16 mM concentrations with 4 mM Ca^2+^ in DMEM media. The complete DMEM medium in each well was replaced with 10%-FBS-supplemented medium containing either free AZ628, nanocarriers alone, or AZ628-loaded nanocarriers. After that, the treated 24-well plates were kept inside the incubator for 48 h until the MTT assay was performed. The similar treatment was prepared to perform MTT assay at 24 h time point to evaluate time-dependent cytotoxicity in MCF-7 cell line.

### 3.9. MTT (3-(4,5-dimethlthiazol-2-yl)-2,5-diphenyltetrazolium Bromide) Assay in a Different Cell Line

MTT assay was performed for the estimation of cytotoxicity of the free drug, free NPs and drug-loaded NPs in MCF-7 and 4T1 cell lines after 24 h and 48 h of treatment. 50 μL of MTT (5 mg/mL in PBS) was added to each well of the plates which were subsequently incubated for 4 h at 37 °C in the incubator humidified with CO_2_ (5%) to convert into formazan crystals. Next, 300 μL of DMSO solution was added to each well after the removal of the MTT medium. After 5 min, the plates were shaken vigorously on the microplate reader to solubilize the purple colored formazan crystals in DMSO. The absorbance of dissolved formazan crystals was measured spectrophotometrically by using a microplate reader (BIO-RAD-Microplate Reader) at a wavelength of 595 nm with reference to 630 nm. The experiments were performed in triplicate and presented as average ± SD.

### 3.10. Experimental Investigation

The cell viability in percent was quantified by using the values extracted from the MTT assay:$$\%\;Cell\;viability=\frac{TS}{C}\times 100\%$$
where, *TS* = the absorbance of the treated cells and *C* = the absorbance of the control.

The toxicity (%) of the free drugs and drug-loaded NPs in cancer cell line was calculated, respectively, by using the following formula:$$\%\;cytotoxicity=CV_{untreated\;cells}-CV_{free\;AZ628}$$
$$\%\;cytotoxicity=CV_{untreated\;cells}-CV_{NP\;bound-AZ628}$$
*CV_untreated cells_* is the cell viability (%) of the control, *CV_free AZ628_* is the cell viability (%) of free drug and *CV_NP bound-AZ628_* is the cell viability (%) of NPs bound drugs.

The enhanced cytotoxicity (%) of AZ628-loaded NPs was assessed using the following formula:$$\%\;Cytotoxicity\;of\;\alpha -KAMCA\;NPs,\;x=100-\left(\frac{R}{Q}\times 100\%\right)$$
$$\%\;Cytotoxicity\;of\;free\;AZ628,\;y=100-\left(\frac{P}{Q}\times 100\%\right)$$
where, *R* = Absorbance of the sample treated with NPs, *P* = Absorbance of the sample treated with free AZ628, *Q* = Absorbance of the control
% Enhanced Cytotoxicity=(C−D)−(A+B)
*C* = % cell viability of the control (100%), *A* = % cytotoxicity of α-KAMCA NPs, *B* = % cytotoxicity of free AZ628 and *D* = % cell viability of AZ628-loaded α-KAMCA NPs. The experiments were performed in triplicate and presented as average ± SD.

### 3.11. Cellular Uptake in MCF-7 and 4T1 Cell Lines

Free AZ628, AZ628-loaded CA and AZ628-loaded α-KAMCA NPs prepared with 40 µM and 100 µM of the drug. The freshly prepared treatment was used to treat MCF-7 (~5 × 10^6^ cells) and 4T1 cells (~5 × 10^6^) to evaluate the dose-dependent cellular uptake for AZ628. The supernatant of culture media was collected after 4 h of the treatment and centrifuged at 13,000 rpm for 30 min at 4 °C. The resultant supernatant was subjected to HPLC to determine the amount of drugs present in the supernatant (representing the free drug not taken up by the cells).

Finally, the cellular uptake was calculated by using the following formula:Cellular uptake=DC−FC
$$\%Cellular\;uptake=\frac{DC-FC}{DC}\times 100$$
where, *DC* = Initial drug concentration, *FC* = Free drug concentration in the supernatant.

The experiments were performed in triplicate and presented as average ± SD.

### 3.12. Western Blot

MCF-7 cells (~5 × 10^6^ cells) were seeded and incubated at 37 °C in a humidified atmosphere with 5% CO_2_ for 24 h. After that, the seeded cells were treated with CA NPs, α-KAMCA NPs, CMCA NPs, AZ628-loaded CA NPs, AZ628-loaded α-KAMCA NPs, AZ628-loaded CMCA NPs and free AZ628 for 24 h. CMCA NPs were formulated according to the protocol of our previous study [4], however in the current study only CA NPs and α-KAMCA NPs were discussed. After treatment with the free drug, free NPs, and drug-loaded NPs, the cells were incubated for 24 h, followed by the cell detachment and cell lysis at 4 °C by using a lysis reagent (protease inhibitor, phosphatase inhibitor, sodium fluoride & stable stock lysis buffer). The lysed cells were centrifuged at 13,000 rpm for 30 min at 4 °C to collect the proteins from the supernatant. Quick Start Bradford Protein Assay kit (Bio-Rad) was used in estimating the total protein contents in the cell lysates. About 10 μg of total proteins were loaded for SDS-PAGE in stain-free Mini protein TGX gels (Bio-rad, USA) in the 1× running buffer at 50 V for 2 h. Proteins were transferred to a nitrocellulose membrane (Bio-rad, Germany) by wet blotting at 0.35 A for 2 h and the membrane was blocked with 5% skimmed milk in tris-buffered saline along with Tween-20 (1× TBST) for 1 h at room temperature. The membrane was incubated overnight with primary antibodies (T-MAPK, T-Akt, p-MAPK, p-Akt, Caspase-3 and GAPDH as an internal control) at 1:1000 dilutions at 4 °C with gentle shaking. The membrane was washed with TBST (5 × 10 mL) for 5 min each and further incubated with a secondary antibody (Thermo Fisher Scientific, Waltham, MA, USA) at 1:3000 dilution for 1 h at room temperature with gentle shaking. After the incubation with secondary antibody, the membrane was washed with TBST (5 × 10 mL) for 5 min to remove any unbound antibody. The membrane was incubated with a mixture of luminol and peroxide Clarity Western ECL substrate (Bio-rad, USA) at room temperature for 5 min. Protein bands were visualized using Chemidoc XRS Imaging system (Bio-rad, Hercules, CA, USA) following the manufacturer’s protocol.

### 3.13. FE-SEM

α-KAMCA NPs and CA NPs were prepared according to the protocol described above and centrifuged at 13,000 rpm for 30 min and the supernatant was removed. The pellets were re-suspended in 1 mL Milli-Q water prior to centrifugation at 13,000 rpm for 30 min. Finally, 30 µL Milli- Q water was added to the pellets after the removal of the supernatant. The sample was placed on a sample holder coated with carbon tape and left to dry at room temperature. After that, the samples with subjected to platinum sputtering for 40 s with 30 mA sputter current at 2.30 tooling factor. The morphology and size of the samples were visualized at 5.0 kV and 2.0 kV using FE-SEM (Hitachi/SU8010, Tokyo, Japan).

### 3.14. Biodegradability Profiles of α-KAMCA NPs, AZ628-Loaded α-KAMCA NPs and AZ628-Loaded CA NPs

α-KAMCA NPs were generated using 20 mM Ca^2+^ and 20 mM alpha-ketoglutaric acid salt in 200 µl PCR tubes. AZ628 at 25 µM, exogenous calcium at 20 mM and α-ketoglutaric acid salt at 20 mM were added to 200 µl of freshly prepared DMEM solution to formulate highly concentrated AZ628-loaded α-KAMCA NPs. AZ628 was added in 25 µM strength with 20 mM exogenous Ca^2+^ to 200 µL of freshly prepared DMEM solution. All the mixtures were incubated at 37 °C for 30 min to generate the NPs (with or without the loaded drug). After that, the suspended NPs were added to 800 µL of DMEM media at different pH ranging from 7.5 to 5.0. Next, the turbidity of the NPs suspensions of different pH was measured at 320 nm by a UV-VIS spectrophotometer. The experimental results of three replicates were presented as average ± SD.

### 3.15. Stability test of NPs

CA NPs were formulated using 4 mM exogenous Ca^2+^ in 1 mL freshly prepared DMEM media and subjected to incubation at 37 °C for 30 min. Likewise, α-KAMCA NPs were generated by mixing 4 mM α-ketoglutaric acid with 4 mM Ca^2+^ in 1 mL of DMEM media and placed inside an incubator of 37 °C for 30 min. Ten percent FBS was added to each formulation. After the formulation, the turbidity of the NPs was measured at 320 nm wavelength using a UV-visible spectrophotometer at different timepoints, 0, 0.5, 1, 2, 4, 8, 24, 48 and 72 h to check the stability of CA and α-KAMCA NPs. Throughout the process CA and α-KAMCA NPs were kept at 37 °C.

### 3.16. Statistical Analysis

Statistical significance was analyzed in drug-treated versus drug-loaded NPs-treated groups by one-way ANOVA, followed by post-hoc analyses using the Scheffe multiple comparisons test (SPSS version 23 for Windows). The minimal level of statistical significance was *p* < 0.05 with 95% confidence interval (CI). The experiments were performed in triplicate and presented as average ± SD.

## 4. Discussion

Different parameters such as diverse concentration gradients of salts, changed pHs of the media, altered incubation time and temperature, and partial replacement of phosphate with carbonate [20,21] and Ca^2+^ with a divalent cation such as magnesium (Mg^2+^) [40] or strontium (Sr^2+^) [22] have been highly studied to improve the growth kinetics of apatite-based NPs. The uniqueness of these NPs is the pH sensitivity towards a tumor acidic environment and the ability to release the drugs inside the endosomes after internalization [20].

We previously reported pH-sensitive citrate-modified CA (CMCA) and succinate-modified CA (SMCA) NPs for the delivery of Doxorubicin, a hydrophilic anthracycline chemotherapeutics to the human breast cancer cell line [4] with promising drug binding affinity, enhanced cellular uptake, and efficient cytotoxicity. CMCA and SMCA, prepared by modifying CA with three carboxyl groups-containing citrate and two carboxyl groups-containing succinate, respectively, has inspired us to explore the physicochemical properties and drug delivery efficacy of CA NPs modified with α-ketoglutarate which contains one ketone group and two carboxylic groups.

α-KAMCA NPs were prepared by incubating at 37 °C for 30 min in a high glucose medium containing fixed concentrations of inorganic phosphate (PO_4_^3−^), calcium (Ca^2+^) and bicarbonate (HCO_3_) ions, with different concentrations of alpha-ketoglutaric acid salt. The formation of NPs was ensured by performing turbidity analysis (Figure 1A) and optical image analysis (Figure 2). The consistent increase in turbidity for α-KAMCA NPs were noticed with increasing concentration of the α-ketoglutarate salt, which clearly indicates the formation of an increased number of particles with an increase in the amount of ketoglutarate salt added. The optical image analysis demonstrated that the addition of 1–4 mM concentration of ketoglutarate salt accelerated formation of the particles; however, the number was significantly reduced at 8–16 mM concentrations of the salt. The larger particles formed at 8 mM–16 mM concentration of alpha-ketoglutaric acid was correlated with the gradual rise in turbidity of the particles. At higher concentration (8–16 mM), α-ketoglutaric acid might have the tendency to aggregate which was further supported by optical image analysis (Figure 2F,G) and DLS analysis (Figure 4E,F).

The enhanced therapeutic efficacy is an essential prerequisite for the engineered nano-drug delivery system, which can be achieved by increasing the circulation half-life of the drug, reducing drug accumulation in the healthy tissues and by releasing the payload into the targeted tumor site [12,41,42]. The unique size-dependent properties of nanomaterials do not allow them to cross the endothelial tight junctions of the healthy vascular lining, however, the NPs can simply cross the leaky tumor vasculature through the EPR effect which also relies on defective lymphatic drainage for prolonged accumulation in the tumour site [42]. Three principal factors usually determine this multi-step process—the increased relative surface area, nano-scale size and surface charge of the nanomaterials [43,44,45]. Higher EPR effect can be produced with particles of average size of 20 nm to 1 µm by prolonging systemic circulation, avoiding opsonization and reducing the uptake rate of MPS (mononuclear phagocyte system) [43,46,47]. The average diameter for α-KAMCA NPs formulated with 4 mM ketoglutarate salt was found to be within 250–300 nm range (Figure 1C) with monodisperse PDI value 0.322 (Figure 4D) and therefore, would be a strong candidate for passive targeting of tumor based on the EPR effect. The zeta potential for CA and α-KAMCA NPs was between −9 to −13 mV, implying no significant difference in the surface charge between the two different NPs (Figure 1D).

In this work, a hydrophobic drug AZ628 was incorporated into CA NPs and α-KAMCA NPs. The drug-particle interaction was verified by drug loading capacity, surface charge of the resultant complex and its effect on cytotoxicity in different cancer cell lines. The fabricated AZ628-loaded CA NPs and α-KAMCA NPs showed average diameters of approximately 456–591 nm (Figure 3A) and 366–479 nm (Figure 3B), respectively. Drug-loaded NPs showed the concentration-dependent increase in particle size proportionally with the increased drug concentration (Figure 3A,B). The surface charges for AZ628-incorporated apatite-based NPs remained same between −9 to −13 mV (Figure 3C,D).

AZ628 could bind with CA and α-KAMCA NPs by ionic interactions of its protonated amine groups in bicarbonate-buffered media of pH 7.4. Drug binding to the NPs was measured by quantifying the amount of unbound drug through HPLC assay (Figure 8). As shown in the result, AZ628 exhibited dose-independent binding affinity towards CA and α-KAMCA NPs. At 60 µM concentration of the drug (Figure 8B), α-KAMCA revealed 4% more binding affinity than CA NPs.

The dissolution profiles of CA NPs and α-KAMCA NPs were assessed at six different pHs of the same buffered media ranging from pH 7.5 to pH 5.0 (Figure 9). In case of apatite-based nanocarriers, pH-responsive drug release is crucial in the target site to avail the substantial toxicity to the cancer cells while lessening the adverse effects to healthy tissues, and to overcome the multidrug resistance [48,49,50,51]. In this study, drug release was indirectly measured by analyzing pH sensitivity of the NPs at the acidic environment through turbidity test. Under normal physiological conditions α-KAMCA NPs, AZ628-loaded CA NPs and AZ628-loaded α-KAMCA NPs were stable; however, the phosphate and carbonate ions in the apatite structure could readily accept excess H^+^ ion from the endosomal acidic microenvironment, causing the particles to be dissolved [52] and enabling AZ628 to passively diffuse from endosome to cytoplasm with the result of enhanced drug accumulation in the tumor region and improved therapeutic efficacy.

AZ628-loaded α-KAMCA NPs demonstrated excellent MAPK and Akt downregulation in the human breast cancer cell lines, reducing proliferation and finally inducing apoptosis as reflected from the activation of cleaved Caspase-3 (Figure 11). Further, it can be explained by the greater cellular uptake of AZ628-loaded α-KAMCA NPs and AZ628-loaded CA NPs than the free drug (Figure 8). The α-KAMCA-facilitated cellular uptake for AZ628 was almost 57% at 100 µM concentration in both cell lines at 4 h of treatment. α-KAMCA NPs showed 25% and 34% more cellular uptake than free AZ628 at 40 and 100 µM concentration, respectively (Figure 8). The enhanced cellular uptake clearly correlates with the decrease in cell viability (Table 2 and Table 3). Thus, the cytotoxic effect of AZ628-loaded α-KAMCA NPs was found to be notably higher than that of the free AZ628, which could be due to the smaller particle size of α-KAMCA NPs, which provide a larger folded surface area to adsorb the sufficient amount of the drug and promote effective cellular internalization via endocytosis [20]. Additionally, the subsequent release of AZ628 from CA and α-KAMCA NPs as a result of fast dissolution of the internalized NPs at acidic pH (Figure 9) contributed to the enhancement in cytotoxicity.

## 5. Conclusions

AZ628 is a potent Raf-kinase inhibitor in the clinical pipeline for curing several types of cancer. Our results demonstrated that α-KAMCA NPs exhibited an excellent drug encapsulation efficiency, superior cellular drug uptake, desired particle size and folded larger surface morphology with rapid dissolution at acidic pH of the microenvironment, ensuing more significant cytotoxicity than CA NPs and the free drug as well. This study also showed that the enhanced dephosphorylation of MAPK and Akt along with the upregulation of Caspase-3 following treatment with AZ628-loaded NPs might be persuasive to inhibit the uncontrolled cancer cell proliferation and thus improve the patient outcome in breast cancer treatment. Hence, α-KAMCA NPs have emerged as a highly promising tool to effectively deliver AZ628 for treating breast carcinoma and encourages further experiments to be carried out to determine the therapeutic efficacy of the nano-formulation in animal models of breast cancer.

## Figures and Tables

**Figure 1 biomedicines-07-00004-f001:**
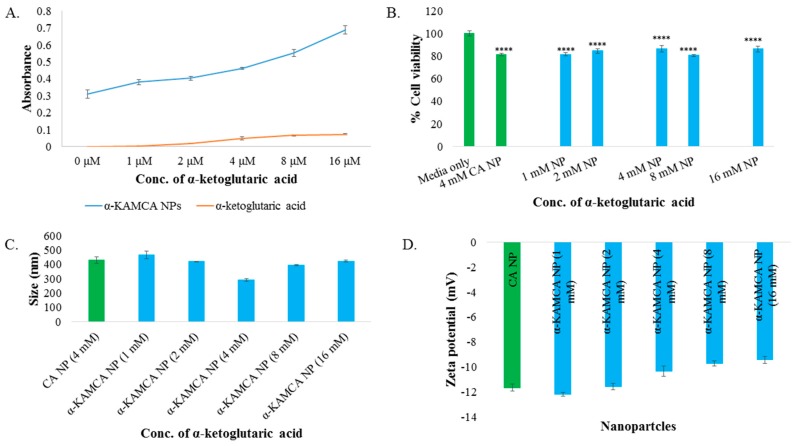
(**A**) Turbidity of α-KAMCA NPs (α-ketoglutaric acid-modified carbonate apatite nanoparticles) and alpha-ketoglutaric acid salt in DMEM media (Dulbecco’s modified eagle medium) measured using different concentrations (0 to 16 mM) of α-ketoglutaric acid salt. (**B**) Cell viability assessment by MTT assay after 48 h incubation of the α-KAMCA NPs in the MCF-7 cell line. Values were significant (****) at *p*-value < 0.0001 vs. untreated cell at CI of 95%. (**C**) Size of α-KAMCA NPs. (**D**) Zeta potential of α-KAMCA NPs.

**Figure 2 biomedicines-07-00004-f002:**
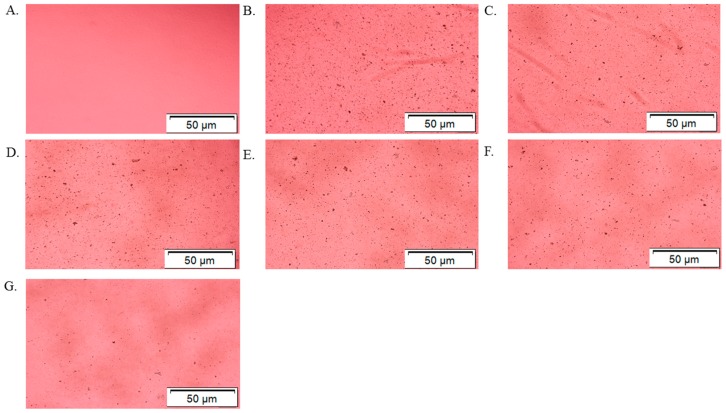
Optical Images of α-KAMCA particle formation. (**A**) only DMEM media without any NPs. (**B**) CA (carbonate apatite) NPs formulated with 4 mM exogenous calcium. (**C**) α-KAMCA NPs formulated with 4 mM exogenous calcium and 1 mM α-ketoglutaric acid. (**D**) α-KAMCA NPs formulated with 4 mM exogenous calcium and 2 mM α-ketoglutaric acid. (**E**) α-KAMCA NPs formulated with 4 mM exogenous calcium and 4 mM α-ketoglutaric acid. (**F**) α-KAMCA NPs formulated with 4 mM exogenous calcium and 8 mM α-ketoglutaric acid. (**G**) α-KAMCA NPs formulated with 4 mM exogenous calcium and 16 mM α-ketoglutaric acid.

**Figure 3 biomedicines-07-00004-f003:**
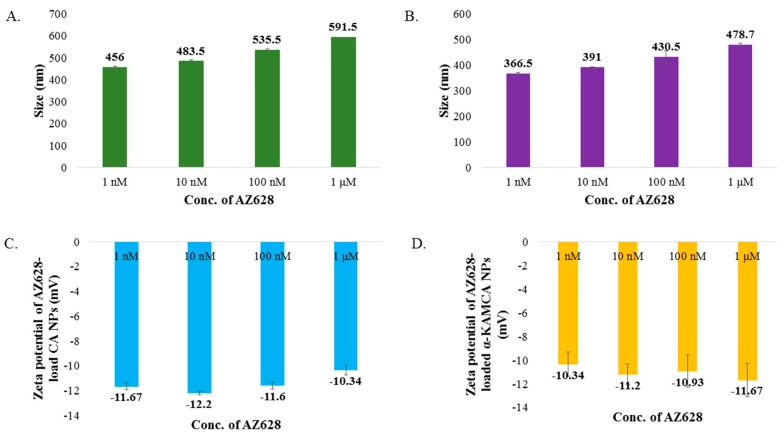
Size and zeta potential of AZ628-loaded CA and α-KAMCA NPs. (**A**) Size of AZ628-loaded CA NPs. (**B**) Size of AZ628-loaded α-KAMCA NPs. (**C**) Zeta potential of AZ628-loaded CA NPs. (**D**) Zeta potential of AZ628-loaded α-KAMCA NPs.

**Figure 4 biomedicines-07-00004-f004:**
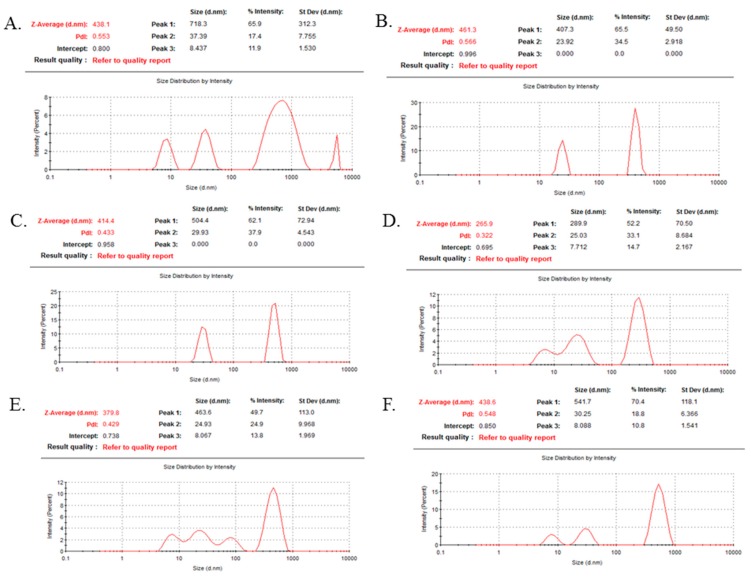
Particle size distribution analysis of CA and α-KAMCA NPs by dynamic light scattering (DLS). (**A**) CA NPs prepared with 4 mM Ca^2+^. (**B**) α-KAMCA NPs prepared with 4 mM Ca^2+^ and 1 mM α-ketoglutaric acid. (**C**) α-KAMCA NPs prepared with 4 mM Ca^2+^ + and 2 mM α-ketoglutaric acid. (**D**) α-KAMCA NPs prepared with 4 mM Ca^2+^ and 4 mM α-ketoglutaric acid. (**E**) α-KAMCA NPs prepared with 4 mM Ca^2+^ and 8 mM α-ketoglutaric acid). (**F**) α-KAMCA NPs prepared with 4 mM Ca^2+^ and 16 mM α-ketoglutaric acid.

**Figure 5 biomedicines-07-00004-f005:**
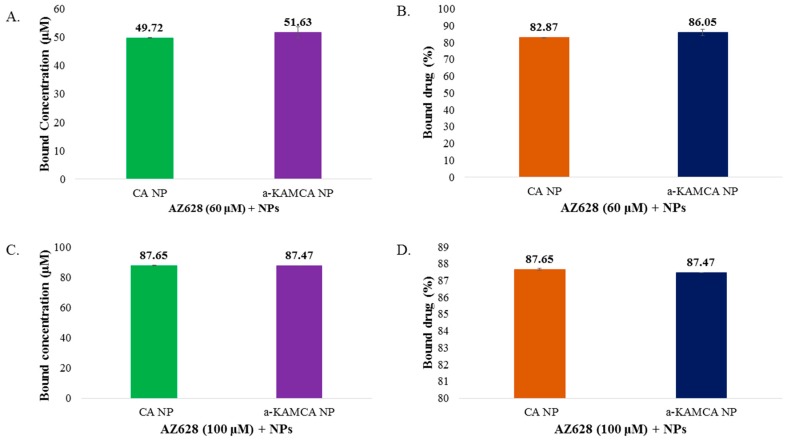
Estimation of AZ628 encapsulation efficacy. (**A**) Drug binding amount (μM) of CA, α-KAMCA NPs NPs for 60 µM AZ628. (**B**) Drug binding efficiency (%) of CA, α-KAMCA NPs NPs for 60 µM AZ628. (**C**) Drug binding amount (μM) of CA, α-KAMCA NPs NPs for 100 µM AZ628. (**D**) Drug binding efficiency (%) of CA, α-KAMCA NPs NPs for 100 µM AZ628.

**Figure 6 biomedicines-07-00004-f006:**
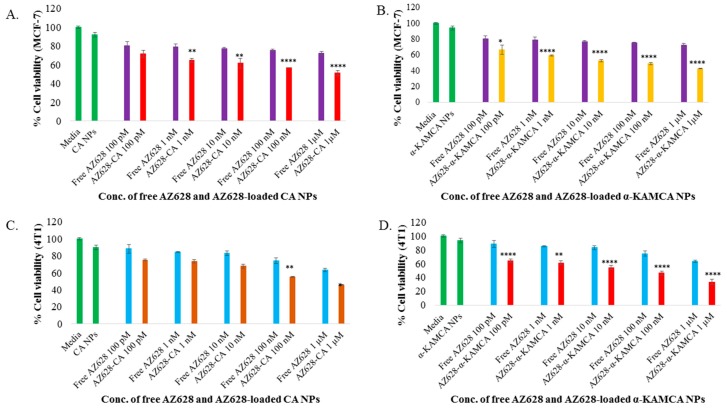
MTT assay of AZ628-loaded CA and α-KAMCA NPs in MCF-7 and 4T1 cell line. (**A**) Cell viability assessment by MTT assay after 48 h incubation of the free AZ628 and AZ628-loaded CA NPs in MCF-7. (**B**) Cell viability assessment by MTT assay after 48 h incubation of the free AZ628 and AZ628-loaded α-KAMCA NPs in MCF-7. (**C**) Cell viability assessment by MTT assay after 48 h incubation of the free AZ628 and AZ628-loaded CA NPs in 4T1. (**D**) Cell viability assessment by MTT assay after 48 h incubation of the free AZ628 and AZ628-loaded α-KAMCA NPs in 4T1. Values were significant (*) at *p*-value 0.01 to 0.05, very significant (**) at *p*-value 0.001 to 0.01, highly significant (***) at *p*-value 0.0001 to 0.001, extremely significant (****) at *p*-value < 0.0001 vs. the same treatment of free AZ628 at CI of 95%.

**Figure 7 biomedicines-07-00004-f007:**
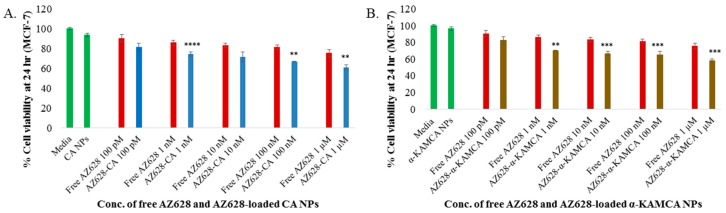
MTT assay of AZ628-loaded CA and α-KAMCA NPs in MCF-7 after 24 h incubation. (**A**) Cell viability assessment by MTT assay of the free AZ628 and AZ628-loaded CA NPs. (**B**) Cell viability assessment by MTT of the free AZ628 and AZ628-loaded α-KAMCA NPs. Values were significant (*) at *p*-value 0.01 to 0.05, very significant (**) at *p*-value 0.001 to 0.01, highly significant (***) at *p*-value 0.0001 to 0.001, extremely significant (****) at *p*-value < 0.0001 vs. the same treatment of free AZ628 at CI of 95%.

**Figure 8 biomedicines-07-00004-f008:**
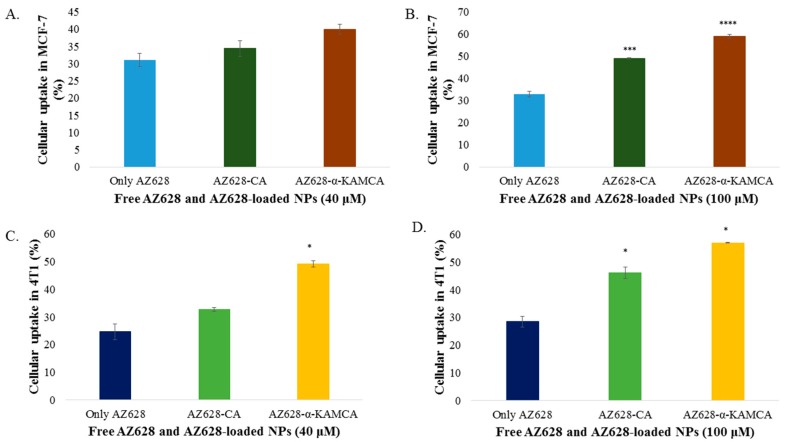
Cellular uptake of free AZ628, AZ628-loaded CA NPs and AZ628- loaded α-KAMCA NPs by MCF-7 and 4T1 cell line. (**A**) Cellular uptake of AZ628, AZ628-loaded CA and α-KAMCA NPs for 40 µM of the drug in MCF-7. (**B**) Cellular uptake of AZ628, AZ628-loaded CA and α-KAMCA NPs for 100 µM of the drug in MCF-7. (**C**) Cellular uptake of AZ628, AZ628-loaded CA and α-KAMCA NPs for 40 µM of the drug in 4T1. (**D**) Cellular uptake of AZ628, AZ628-loaded CA and α-KAMCA NPs for 100 µM of the drug in 4T1. Values were significant (*) at *p*-value 0.01 to 0.05, highly significant (***) at *p*-value 0.0001 to 0.001, extremely significant (****) at *p*-value < 0.0001 vs. the same treatment of free AZ628 at CI of 95%.

**Figure 9 biomedicines-07-00004-f009:**
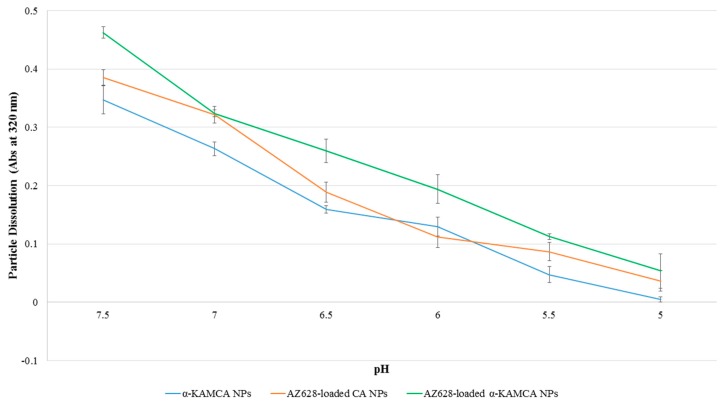
pH-dependent dissolution of CA, α-KAMCA and AZ628-loaded α-KAMCA NPs.

**Figure 10 biomedicines-07-00004-f010:**
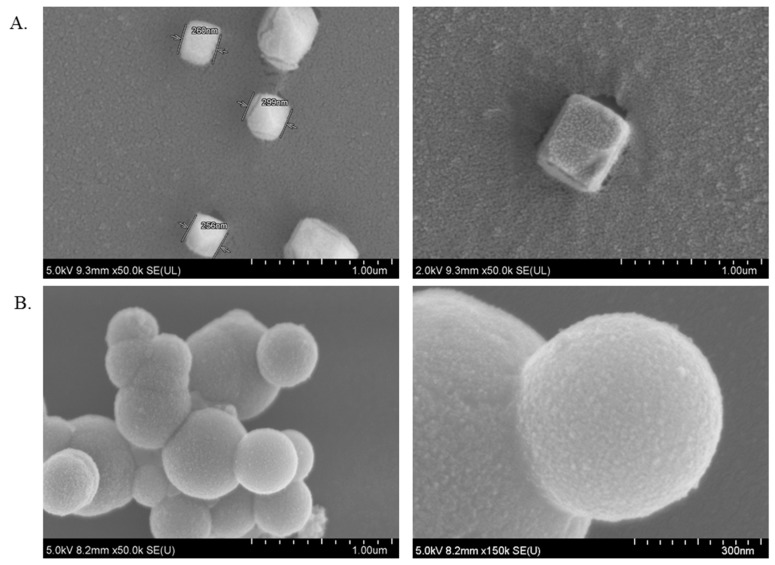
Shape and size characterization by FE-SEM. (**A**) Shape, size and surface characterization of α-KAMCA NPs. (**B**) Shape, size and surface characterization of CA NPs.

**Figure 11 biomedicines-07-00004-f011:**
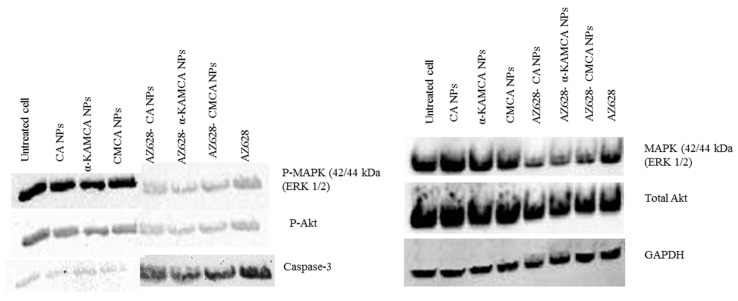
Western blot analysis for AZ628, AZ628-loaded CA NPs, AZ628-loaded CMCA (citrate-modified carbonate apatite) NPs and AZ628-loaded α-KAMCA NPs.

**Figure 12 biomedicines-07-00004-f012:**
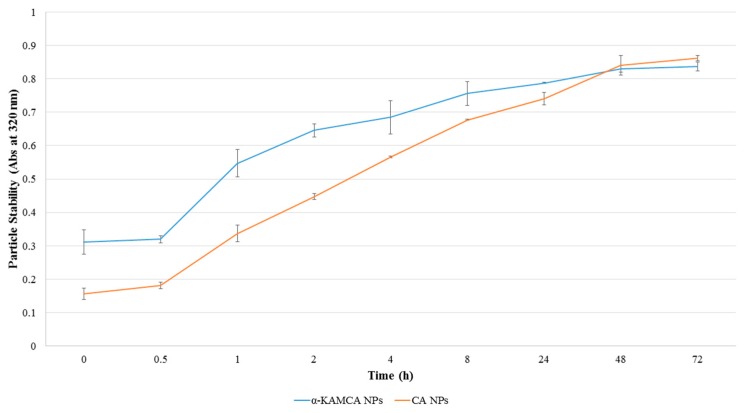
Time-dependent stability test of CA and α-KAMCA NPs.

**Table 1 biomedicines-07-00004-t001:** The encapsulation efficiency and drug loading capacity of CA (carbonate apatite) NPs (nanoparticles) and α-KAMCA (α-ketoglutaric acid-modified CA) NPs.

NPs	Initial Concentration of AZ628	% Drug Loading (*w*/*w*)
CA	60 μM (27.09 μg/mL)	14.97 ± 0.1
CA	100 μM (45.15 μg/mL)	26.96 ± 0.89
α-KAMCA	60 μM (27.09 μg/mL)	18.68 ± 0.05
α-KAMCA	100 μM (45.15 μg/mL)	32.28 ± 0.04

**Table 2 biomedicines-07-00004-t002:** Enhancement of cytotoxicity (%) for AZ628-loaded α-KAMCA nanoparticles (NPs).

Concentration of AZ628	MCF-7	4T1
100 pM	8.07 ± 5.78	17.35 ± 2.27
1 nM	13.7 ± 0.71	16.67 ± 3.01
10 nM	18.59 ± 1.52	22.02 ± 2.97
100 nM	20.58 ± 1.53	21.35 ± 2.61
1 µM	21.02 ± 0.57	23.21 ± 3.67

**Table 3 biomedicines-07-00004-t003:** Enhancement of cytotoxicity (%) for AZ628-loaded CA NPs.

Concentration of AZ628	MCF-7	4T1
100 pM	0.88 ± 3.58	3.01 ± 1.08
1 nM	6.63 ± 1.99	1.22 ± 0.24
10 nM	7.67 ± 5.1	5.02 ± 2.38
100 nM	10.51 ± 0.19	9.19 ± 0.68
1 µM	10.58 ± 2.81	7.48 ± 0.36

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
