# Peer review of "α-Ketoglutaric Acid-Modified Carbonate Apatite Enhances Cellular Uptake and Cytotoxicity of a Raf-Kinase Inhibitor in Breast Cancer Cells through Inhibition of MAPK and PI-3 Kinase Pathways"

_biomedicines, 2019, doi:10.3390/biomedicines7010004_

Reviewer 1 Report

My comments and suggested changes are enlisted below:

1. The CA & α-KAMCA NPs preparation part is poorly written and more details are needed.

2. Authors mentioned the drug encapsulation efficiency, but what was the drug loading? This is important. 

3. Detailed description of the Malvern Nano Zetasizer method should be provided. Is it size/intensity or number measurement? Also, the DLS size figure should be provided.

4. Electron microscopy images of NPs (SEM or TEM) need to be provided to confirm the particle size.

5. Scale bars are not visible in Figure 2.

6. What was the positive control in cell viability assays (Figure 5)? Also, how long the assay was performed. Did author do any time-dependent viability assay using these NPs?

7. In Figure 8, authors mentioned the shape, but, no discussion has been provided in the text.

8. No data has been provided for the drug release profile? These details are important and needs to add in the manuscript.

9. What was the rationale behind selecting MCF-7 and 4T1 cell lines?

10. The method section is poorly written (section 2). Each experiment should be included with how the samples are prepared and measured using what instrument, company name, material supplier, etc.

11. What about the stability of NPs? Was any study performed? Why not?  

Author Response

Answer:

1. The Method section has been re-written. The detailed preparation of CA NPs and α-KAMCA NPs has been added.2. Encapsulation efficiency (EE) gives an idea about % of the drug that is successfully entrapped/adsorbed into nanoparticles. It is calculated as follows: %EE = [(Total drug added – Free or unentrapped drug)/Total drug added] X 100. We can use either the direct method by estimating the drugs encapsulated into the particles or indirectly by measuring what was not encapsulated into the nanoparticles. Since we included the appropriate control by including free drugs in DMEM and collected the supernatant after centrifugation as in case of the sample, the indirect method is also reliable.

2. Measuring drug loading efficacy by estimating the amount of drug loaded into a certain amount of lyophilized powder of the formulation is technically difficult for us considering the high cost of the drug.   

3. The Method section has been re-written. The DLS curve of nanoparticles has been added to the text. The size distribution of the nanoparticles was similar in z-average measurement, DLS curve and FE-SEM analysis. There was no overestimation of nanoparticle size.

4. FE-SEM images were provided in the manuscript. In page 17, Figure 10.

5. In figure 2, the size of scale bar has been increased.

6. In figure 5 (current figure 6), the positive control is the media where the cells has been treated with only DMEM solution.

We have tried different time points for cell proliferation studies (4 h, 24 h, 48 h and 72 h); however, the best result was obtained in 48 h time point, which might be correlated to the half-life of the drug. To address the reviewer’s comment, we also added 24 h treatment results; however at lower concentration the cytotoxicity is not that significant.

7. In Figure 8 (current Figure 10), the shape of the NPs has been discussed.

8. The pH dependent drug release profile has been discussed.

9. Human breast cancer cell line MCF-7 was used in vitro breast cancer experiment due to its hormone sensitivity and oestrogen receptor (ER) expression [36]. The murine mammary adenocarcinoma 4T1 cell line was also selected to mimic the breast cancer model for its resemblance to human triple negative breast cancer [37] and also an animal model for stage IV human breast cancer.

Reference –

36. Holliday, D.L.; Speirs, V. Choosing the right cell line for breast cancer research. Breast cancer research : BCR 2011, 13, 215, doi:10.1186/bcr2889.

37. Silva, V.L.; Ferreira, D.; Nobrega, F.L.; Martins, I.M.; Kluskens, L.D.; Rodrigues, L.R. Selection of Novel Peptides Homing the 4T1 CELL Line: Exploring Alternative Targets for Triple Negative Breast Cancer. PloS one 2016, 11, e0161290, doi:10.1371/journal.pone.0161290.

10. The Method section has been re-written. The company name and supplier name has been added with the instrument.

11. The stability test has been added.

Reviewer 2 Report

The authors have presented a novel approach for the delivery of AZ628 using nanoparticles targeting breast cancer cell lines. The nanoparticles are fabricated using Carbonate Apatite (CA) along with α-Ketoglutaric acid to increase the surface area and targeted delivery. The study includes characterization of the nanoparticles, effect of particle delivery based on different pH and cytotoxicity assays for the cell lines under investigation. The authors have demonstrated similar work before using citrate and succinate as the modifiers for the nanoparticles. However, the encapsulation of hydrophobic drug (AZ628) within CA particles can be an attractive method for targeted delivery in breast cancer therapy. However, the manuscript can be accepted after dealing with the minor issues mentioned as follows:

1. Under the 3. Results section, subheading 3.2 Size and Surface Charge, the particle sizes of CA NPs are given in value with approximation, however I would suggest the authors to have an uncertainty value with the particle size.

2. In Figure 7, the x axis title should be “pH” instead of “different pH”. 

3. In 41 line of 4. Discussion section, “Fig 7” should be replaced with “Figure 7”.

Author Response

Answer:

1. The particle sizes of NPs has been presented as average ± standard deviation to explain the size range. The DLS curve of nanoparticles has been added to the text. The size distribution of the nanoparticles was similar in z-average measurement, DLS curve and FE-SEM analysis. There were no overestimation of nanoparticle size.

2. In Figure 7 (current Figure 9), the x-axis has been changed to pH.

3. It has been corrected.

Reviewer 3 Report

The manuscript titled "α-Ketoglutaric Acid-Modified Carbonate Apatite Enhances Cellular Uptake and Cytotoxicity of a Raf-kinase Inhibitor in Breast Cancer Cells through inhibition of MAPK and PI-3 kinase pathways" by Sultana Hossain and co-workers deals with characterisation and use of Raf-kinase Inhibitor-loaded Carbonate Apatite nanoparticles to target Breast Cancer Cells. The paper is well structured and written in a good English language. The authors demonstrated with their results that α-Ketoglutaric Acid can be envisaged as a promising therapeutic tool to treat breast cancer. The paper in this current form is not yet ready to be published but it requires several revisions. The suggested\recommended changes\modifications are listed in the following points:

1) English: a moderate mother tongue revision of manuscript is suggested in order to remove typos and improve its readability

2) Cells lines: what was the reason of choice specifically these two cell lines (MCF-7 and 4T1)? Why a control healthy cell line was not designated? Please comment\justify.

3) TEM and SEM images of NPs  in order to confirm size\morphology are needed to be added to the paper: please provide them, if possible. Optical images of Figure 2 are rather poor. Moreover, in Figure 2 scale bars are not properly visible. Correct them.

4) Following point 3 a quantification of NPs size is helpful for readers comprehension

5) What about release profiles? These data are missing. Discuss \provide them

6) Figure 3-4: data presentation needs to be improved by adding labels\arrows to explain better  the measurements

7) In vivo experiments: are planned\designed by authors? Please comment\add an outlook

8) Figure 10 is missing: please provide it and add\update comments\discussion

9) A comprehensive NPs uptake\internalisation\co-localisation investigation\analysis\quantification is totally overlooked. Please provide it. 

Author Response

Answer:

1. The manuscript has been checked carefully and corrected the typos.

2. Human breast cancer cell line MCF-7 was used to model in vitro breast cancer experiment due to its hormone sensitivity and oestrogen receptor (ER) expression [36]. The murine mammary adenocarcinoma 4T1 cell line was also selected to mimic the breast cancer model for its resemblance to human triple negative breast cancer [37] and also an animal model for stage IV human breast cancer.

Reference –

36. Holliday, D.L.; Speirs, V. Choosing the right cell line for breast cancer research. Breast cancer research : BCR 2011, 13, 215, doi:10.1186/bcr2889.

37. Silva, V.L.; Ferreira, D.; Nobrega, F.L.; Martins, I.M.; Kluskens, L.D.; Rodrigues, L.R. Selection of Novel Peptides Homing the 4T1 CELL Line: Exploring Alternative Targets for Triple Negative Breast Cancer. PloS one 2016, 11, e0161290, doi:10.1371/journal.pone.0161290.

We wanted to focus on cancerous cell and the cytotoxic effect of NPs on cell proliferation. However, we found no cytotoxicity on cancerous cell line for our NPs.

3. FE-SEM images were provided in the manuscript and have been discussed in terms of size and shape of NPs. In page 17, Figure 10.

In Figure 2, the size of scale bar has been increased.

4. The DLS curve of nanoparticles has been added to the text. The size distribution of the nanoparticles was similar in z-average measurement, DLS curve and FE-SEM analysis. There were no overestimation of nanoparticle size.

5. The pH dependent drug release profile has been discussed.

6. The comment was not clear to us. However, in Figure 3 and Figure 4 (current figure 5) the measurement value of the bar graph has been labelled.

7. In conclusion section, the further planning for in-vivo experiment was discussed.

 Conclusions

AZ628 is a potent Raf-kinase inhibitor in the clinical pipeline for curing several types of cancer. Our results demonstrated that α-KAMCA NPs exhibited excellent drug encapsulation efficiency, superior cellular drug uptake, desired particle size and folded larger surface morphology with rapid dissolution at acidic pH of the microenvironment, ensuing more significant cytotoxicity than CA NPs and free drug as well. This study also showed that enhanced dephosphorylation of MAPK and Akt along with upregulation of caspase-3 following treatment with AZ628-loaded NPs might be persuasive to inhibit uncontrolled cancer cell proliferation and thus improve patient outcome in breast cancer treatment. Hence, α-KAMCA NPs have emerged as a highly promising tool to effectively deliver AZ628 for treating breast carcinoma, encouraging further experiments to be carried out to determine its therapeutic efficacy in animal models of breast cancer.

8. The manuscript has been checked carefully and corrected the typos.

9. The time-dependent and concentration-dependent cellular uptake study was provided. (Figure – 8).

Round  2

Reviewer 1 Report

The authors responded most of my questions except I am not fully satisfied with their response to my comments 2 and 6.

Comment 2: They can easily measure the drug loading using the same approach they did for encapsulation efficiency, except they need to calculate based on the amount of formulation used.

Comment 6: Author said that in figure 5 (current figure 6), the positive control is the media where the cells have been treated with only DMEM solution. Treatment with only media should be negative control and not positive control?

Author Response

Answer:

1. The drug loading capacity of NPs has been calculated and added to the manuscript.

2. DMEM media treated cells are the negative control whereas nanoparticles treated and different concentration of free drug treated cells are the positive control. The cells were treated with nanoparticles to check the partial toxicity of the nanoparticles. However, from the result it has been found that nanoparticles are not apparently cytotoxic. Likewise, the cells were also treated with the free drugs to evaluate the difference in cytotoxicity for the drug-loaded nanoparticles comparing to the free drug of same concentration. Apart from this, the enhancement of cytotoxicity was measured by using the following formula –% Enhanced Cytotoxicity=(C-D)-(A+B), C = % cell viability of the control (100%), A = % cytotoxicity of α-KAMCA NPs, B = % cytotoxicity of free AZ628 and D = % cell viability of AZ628-loaded α-KAMCA NPs.

Reviewer 3 Report

The authors have carefully addressed all the issues raised by previious revision and manuscript has reached standard quality to be accepted in current form 

Author Response

To improve the quality of the manuscript, statistical significance was analysed in drug-treated versus drug-loaded NPs-treated groups by one-way ANOVA followed by post-hoc analyses.